# Comprehensive Overview of Sleep Disorders in Patients with Chronic Liver Disease

**DOI:** 10.3390/brainsci11020142

**Published:** 2021-01-22

**Authors:** Oana-Mihaela Plotogea, Madalina Ilie, Simona Bungau, Alexandru Laurentiu Chiotoroiu, Ana Maria Alexandra Stanescu, Camelia Cristina Diaconu

**Affiliations:** 1Department 5, “Carol Davila” University of Medicine and Pharmacy, 050474 Bucharest, Romania; drmadalina@gmail.com; 2Department of Gastroenterology, Clinical Emergency Hospital of Bucharest, 105402 Bucharest, Romania; 3Department of Pharmacy, Faculty of Medicine and Pharmacy, University of Oradea, 410028 Oradea, Romania; simonabungau@gmail.com; 4Department of Surgery, Clinical Emergency Hospital of Bucharest, 105402 Bucharest, Romania; chiotoroiu@yahoo.com; 5Department of Family Medicine, “Carol Davila” University of Medicine and Pharmacy, 050474 Bucharest, Romania; alexandrazotta@yahoo.com; 6Department of Internal Medicine, Clinical Emergency Hospital of Bucharest, 105402 Bucharest, Romania

**Keywords:** sleep disorders, cirrhosis, hepatic encephalopathy, melatonin, questionnaires, actigraphy, therapies

## Abstract

The impact of sleep disorders (SDs) on patients with chronic liver diseases (CLD) is tremendous. SDs are frequently encountered among these patients and interfere with their quality of life. This review aims to present the data available so far about the prevalence, phenotypes, and proposed pathophysiological mechanisms of SDs in CLD. Moreover, we proposed to search the literature regarding the most reliable methods to assess SDs and the possible therapeutic options in patients with CLD. The main results of this review show that when it comes to prevalence, the percentages reported vary widely between studies performed among populations from the USA or Europe and those coming from Asian countries. Furthermore, it has been proven that SDs may also be present in the absence of neurocognitive disorders attributable to hepatic encephalopathy (HE), which contradicts traditional suppositions where SDs were considered part of the clinical scenario of HE. Currently, there are no specific recommendations or protocols to assess SDs in CLD patients and data about the therapeutic management are limited. Taking into consideration their impact, a protocol for diagnosing and managing SDs should be developed and included in the daily practice of hepatologists.

## 1. Introduction

The attention of medical community is focused especially on the medical treatment of liver diseases and their complications, in order to increase survival and prevent complications. Less importance is given to the quality of life of these patients and the identification of the main causes responsible for its reduction.

Sleep disorders/disturbances (SDs) are strongly associated with psychological problems (e.g., anxiety and depression), leading to impaired quality of life [1]. SDs are often found in patients with chronic liver disease (CLD), such as steatosis, steatohepatitis, and especially cirrhosis, even in the absence of neuropsychiatric impairment [2].

Traditionally, hepatic encephalopathy (HE) has been incriminated as the main cause for sleep disorders, but recent studies [2,3,4,5,6] have reported sleep abnormalities in patients without neurocognitive disorders attributable to HE. Among the possible incriminated mechanisms have been mentioned imbalances of melatonin metabolism, imbalances of thermoregulation, and increased level of serum ammonia that crosses the blood-brain barrier. Thus, the associated causal factors are in continuous research, the currently known data being insufficient to understand their occurrence.

Regarding the assessment of SDs in this population, researchers have suggested various subjective and/or objective tools, which are largely used also in other medical conditions. So far, no screening protocol has been implemented in the daily clinical practice for investigating SDs among patients with CLD.

This article presents a comprehensive overview regarding the prevalence, classification and phenotypes of SDs together with their mechanisms, assessment, and management options in patients with CLD, especially liver cirrhosis. Electronic databases (PubMed Central, Cochrane Library, Embase—EMBASE for Excerpta Medica dataBASE) were searched, using the following keywords: “sleep”, “sleep disorders”, “sleep disturbances”, “liver”, “chronic liver disease”, “cirrhosis”, “hepatitis”, and “nonalcoholic fatty liver disease”. We included in this systematic review 163 publications following Preferred Reporting Items for Systematic Reviews and Meta-Analyses (PRISMA) guidelines (Figure 1). The inclusion criteria were human and animal studies where sleep disorders and liver disease coexisted. All relevant articles in English or with English abstracts were retrieved. In addition, cross-references from the identified articles were reviewed. Studies referred to pediatric population or sleep disorders encountered in other diseases were excluded, in the absence of hepatic involvement.

## 2. Classification of SDs

SDs are characterized by disturbances in the quality and/or amount of sleep and can manifest as sleep deprivation, sleep fragmentation, and events occurring during sleep [7].

According to the International Classification of Sleep Disorders (ICSD) proposed by the American Academy of Sleep Medicine [8], there are seven major groups that include: insomnia, sleep-related breathing disorders (e.g., obstructive sleep apnea—OSA, central sleep apnea—CSA), circadian rhythm sleep-wake disorders, parasomnias, sleep-related movement disorders, central disorders of hypersomnolence and other sleep disorders (Table 1).

## 3. Epidemiology and Phenotypes of SDs in CLD

In clinical practice, SDs are difficult to estimate in the general population. Their prevalence depends on various factors such as environmental and social factors, individual factors (age, gender, genetics, race) and especially comorbidities [20]. It has been estimated that up to 70 million people in the US and 45 million in Europe face a chronic sleep disorder, with impact on health and daily functioning [7]. Prospective studies have shown that modifications in sleep quality and/or duration are associated with a variety of chronic diseases (metabolic syndrome, hypertension, diabetes, cardiovascular diseases) and with increased all-cause mortality [21,22,23,24,25]. In a meta-analysis, Mathias et al. reported that 44% of adults with chronic pain, irrespective of its cause, experience SDs [26].

In the scientific literature, there are only a few epidemiological studies that estimate the prevalence of SDs in patients with CLD, most of them being conducted among cirrhotics [2,4,27,28,29]. The percentages vary widely among studies due to the heterogeneity of the assessment tools and the differences between the populations studied. Cordoba et al. [27] reported that sleep disturbances occurred in 47% of the patients with cirrhosis, while another study [28] revealed that 69% of cirrhotics complained of disturbed sleep and also of depression. In patients with end-stage liver disease, the prevalence of SDs is 81%, commonly attributed to the presence of hepatic encephalopathy [29]. Thus, there are many factors associated to the liver disease which might influence the prevalence of SDs.

The association between SDs and liver disease was described for the first time in 1954 by Sherlock et al., when they reported the presence of so-called sleep-wake inversion, translated into restless nights and excessive daytime sleepiness, in patients with overt HE [30]. Since then, several researchers investigated the symptoms and signs of SDs in relation with or independent of the presence of HE [5,6,29,31,32,33,34,35].

The most encountered phenotypes of SDs among patients with CLD are insomnia, excessive daytime sleepiness (EDS), obstructive sleep apnea, and restless leg syndrome (Figure 1) [4,31,33].

Insomnia is mainly a clinical diagnosis based on data obtained from the patient and defined as the difficulty of initiating and/or maintaining sleep [33]. Patients with cirrhosis describe insomnia in relation with the impossibility to fall asleep, fragmented sleep, multiple awakenings during the night resulting in poor sleep quality, and impaired daytime functioning [34,35]. Observational studies among patients with cirrhosis showed that insomnia is present in 42–65% of the patients [2,27,28,31,35]. First, Cordoba et al. [27] and later Montagnese et al. [2] showed that insomnia was present even in well-compensated stages of cirrhosis and there is no association between insomnia and neuropsychiatric impairment. In a cross-sectional study of 200 patients with stable liver cirrhosis, Al-Jahdali et al. [35] reported a significantly higher prevalence of insomnia in patients with hepatitis C compared to hepatitis B and other causes of hepatitis. These authors also observed an inverse relationship between the severity of cirrhosis evaluated by Child–Pugh score and prevalence of insomnia. In addition, insomniac cirrhotics tend to be older than non-insomniacs and also have a significantly larger neck size. Interestingly, no correlation was found between body mass index and insomnia [29,35].

Excessive daytime sleepiness (EDS), along with insomnia, is one of the main complaints of patients with cirrhosis and SDs and is encountered in 21–50% of patients [4,6,31,34,36]. Samanta et al. [34] conducted a cross-sectional study including 100 patients with cirrhosis, divided in two groups depending on the presence or absence of HE. Of those patients, 38% presented EDS and 89.5% of them also associated overt HE. A strong correlation between EDS and the presence of HE was also reported by other researchers [2,27,37]. Observational studies showed that there is a significant relationship between EDS and severe cirrhosis. According to Montagnese et al. [2], a link between EDS and neuropsychiatric impairment seen in HE cannot be excluded.

Regarding sleep-related breathing disorders, the most studied in relation with liver disease is obstructive sleep apnea (OSA). This clinical condition is caused by complete or partial obstruction of the upper airways, resulting in chronic intermittent hypoxia [38]. There is a well-established bidirectional correlation between OSA and non-alcoholic fatty liver disease (NAFLD). OSA is seen in 35–45% of patients with obesity and NAFLD, as the two conditions often coexist and share common metabolic alterations and comorbidities (hypertension, type 2 diabetes mellitus, cardiovascular disease) [39]. Reversely, Chou and colleagues [40] showed in a large population-based cohort study an increased incidence of liver disease among people with OSA compared with control group. Researchers reported that in addition to NAFLD, also patients with cirrhosis and viral hepatitis (B and C) showed 3- to 5-fold higher incidences of OSA than the control groups [40].

Another sleep disorder that has been studied in patients with CLD is restless leg syndrome (RLS) or Ekbom’s syndrome, which manifests as the urge to move legs or other extremities during rest. It may also associate unpleasant or painful sensations with diurnal variation and release by movement [41]. Its link with CLD was first described in 2008, by Franco et al., in 62% of the patients included in their study, compared to only 10% of the general population [42]. Interestingly, in a study performed among Japanese patients with CLD, the prevalence of RLS was lower, 16.8% respectively, and almost half of them (48%) also associated other SDs [43]. In a prospective observational study, Rajender and colleagues determined the prevalence of RLS (26%) among cirrhotic subjects and proved to be significantly higher for alcoholic etiology and more severe cirrhosis [44]. Unlike these findings, a larger prospective study reported no differences of RLS prevalence and severity between compensated and decompensated cirrhosis, regardless of its cause [45].

The possible explanation for the great discrepancy between studies regarding the prevalence of RLS comes from the differences in the populations of patients with chronic liver disease (CLD) and restless legs syndrome (RLS). A higher prevalence is seen in studies performed in the USA or Europe compared to Asia. Studies have shown that racial differences exist in not only idiopathic RLS, but also secondary RLS [43].

The most frequent phenotypes of SDs in CLD and their prevalence are presented in Figure 2.

## 4. Discussion

### 4.1. Etiology of CLD

CLD affects more than 30 million people all over the world and its prevalence is continuously rising because of the rapidly increasing number of patients with metabolic syndrome. The main etiologies of CLD are alcohol consumption, viral hepatitis B and C, NAFLD, and genetic diseases (Table 2). The natural course of any CLD is the development of chronic inflammation, fibrosis, and progression to cirrhosis and hepatocellular carcinoma [46].

Irrespectively of the etiology, CLDs are associated with SDs in a large number of patients. Nevertheless, SDs are multifactorial, and their underlying mechanisms are complex and not fully understood [4].

### 4.2. Circadian Clock Misalignment in CLD

In order to understand SDs pathophysiology, the normal characteristics of sleep have been studied. Human sleep encompasses two stages: rapid eye movement sleep (REM) and non-rapid eye movement sleep (NREM), which alternate during sleep. Sleep begins with NREM, divided in four stages, which have synchronous cortical EEG (electroencephalogram) pattern/slow waves sleep (SWS). NREM is usually associated with minimal mental activity and accounts for 80% of sleep. SWS is present mainly in the first third of the night and is correlated with the initiation of sleep and the period of time awake (meaning sleep homeostasis). Afterwards, sleep progresses into REM stage, defined by EEG activation, muscle atonia, and episodes of rapid eye movements. REM accounts for 20% of sleep, being also predominant in the last third of night and related to circadian rhythm and body temperature [47].

In addition to sleep homeostasis, another physiological pathway for sleep regulation is the circadian rhythm (Figure 3).

In humans, the circadian clock system is 24-h timed and controls sleep-wake cycle, eating, fasting, and hormonal secretion. It is made of a central clock represented by the suprachiasmatic nucleus (SCN) located in the hypothalamus and peripheral clock located in organs like liver, kidneys, muscles, heart, and gut [33]. These peripheral tissues are regulated by the SCN through neural and endocrine pathways and have essential roles in maintaining overall homeostasis and controlling metabolism of nutrients. In normal conditions, the circadian clock is set daily by light and food. On one hand, the light reaches retina and via retinal-hypothalamic tract triggers SCN. On the other hand, food directly activates the peripheral clocks [48]. In response to SCN controlling signals the pineal gland synthetizes melatonin which will be increased during sleep and suppressed during daytime, as a consequence of exposure to light. Melatonin is considered a “neuroendocrine transductor” of light-dark cycle and is excreted through urine after being metabolized in the liver [49]. SCN is also responsible for regulation of the cortisol secretion and core body temperature. The 24-h sleep-wake cycle is influenced by glucocorticoid hormones, known as essential messengers. Their action upon SCN is dual, involving serotonin depletion and reducing arginine vasopressin signaling in the SCN. Glucocorticoid treatment administered in numerous inflammatory diseases may cause sleep architecture abnormalities (increased REM latency, increased time spent awake, and SWS). Similarly to glucocorticoids, statins may lead to disruption in the SCN signaling via metabolic impairments, such as release of muscle metabolites, namely prostaglandins [50].

Pathophysiological mechanisms regarding disruption of circadian system have been proposed in order to explain SDs manifestations in patients with CLD. Several studies have evaluated the urinary and plasma levels of melatonin in cirrhosis, assuming that its clearance would be compromised [33,51,52,53,54]. In 1995, Steindl and colleagues [54] reported that in cirrhosis, with subclinical HE, the onset of plasma melatonin increase and nocturnal peak level were displaced to later hours. In accordance with Steindl’s results, Velissaris et al. [52] obtained the same results after they conducted a case-control study showing that patients with cirrhosis and subclinical HE exhibit abnormal melatonin patterns, such as high daytime plasma levels of melatonin, delayed melatonin increase onset, and delayed melatonin peak during sleep. Cordoba et al. [27] suggested that patients with compensated cirrhosis have a characteristic sleep pattern consistent with a shift towards later bedtime and later awakenings.

Later, Montagnese and colleagues [49] assessed simultaneously plasma melatonin and urinary metabolites levels. Their results showed that cirrhotic patients exhibit delays in plasma melatonin rhythm onset and that circadian patterns of urinary metabolites (6-sulfatoxymelatonin (6-HMS)) did not differ in cirrhosis compared to healthy controls [49]. On the contrary, Steindl et al. [54] observed that patients with cirrhosis grade A and B had a significant reduction of 6-HMS during night. In another study, patients with overt HE had both a diurnal and nocturnal increase, associated with a drop of 6-HMS in urine. Thus, advanced liver failure causes reduced metabolism of melatonin in liver and increased leakage to systemic circulation from the portal vein [51]. In patients with portal hypertension, measurable levels of melatonin in the ascitic fluid have also been described [55].

In addition, there are studies showing that patients with cirrhosis also manifest delays in plasma cortisol 24 h rhythm, more exactly in the onset and acrophase, compared with healthy volunteers [33,49]. In any case, there is no significant difference in cortisol rhythm parameters between compensated and decompensated cirrhosis; there is, though, a correlation between acrophase time of plasma cortisol and Child–Pugh score [49]. The rising phases of both plasma melatonin and cortisol profiles indicate the central origin of circadian disruption [49].

However, after assessing sleep disorders, researchers did not find clear links between sleep patterns in cirrhotic patients and melatonin rhythm [49,54,56,57]. In an observational study, the authors concluded that “melatonin profile abnormalities do not explain the sleep–wake disturbances exhibited by these patients” (Montagnese et al., 2010, p. 1781); therefore, alternative pathophysiological mechanisms should be further investigated [49]. However, interestingly, patients with high diurnal and nocturnal melatonin concentrations present daytime sleepiness and fatigue [27,33,55]. Moreover, studies did not demonstrate any correlation neither between laboratory tests (e.g., liver enzymes, bilirubin, albumin) and melatonin level, nor between ammonia and melatonin level [55].

The relation between circadian rhythm and portal hypertension is not completely understood. Experimental studies [58,59,60,61,62] suggested that portacaval shunts disrupt circadian motor activity and melatonin rhythms in rats with HE. There were reported significant correlations between the severity of cirrhosis, the degree of HE, and melatonin level, which gave rise to this hypothesis that circadian rhythm abnormalities are also the effects of neurotoxins on SCN (suprachiasmatic nucleus) and on its afferent/efferent pathways [52,53,56]. Moreover, the same metabolic imbalance that causes HE could affect the circadian system.

In addition to ammonia, there are other neurotoxic substances encountered in HE, like methionine and mercaptans, which disturb the normal enzymatic processes in the brain, leading to synthesis of false neurotransmitters [51]. Ammonium-containing diets are proven to have toxic effects on the pineal gland and central neurotransmission. At the same time, hyperammonemia correlates with EEG modifications [60], with subsequent unrestful sleep, and excessive daytime sleepiness [37,62]. Even patients with minimal HE exhibit a high percentage of REM sleep with the absence of SWS (slow waves sleep) [56]. The architecture of sleep phases is thought to be disrupted due to disturbances in the homeostatic regulation of sleep, in adenosine level in particular [63]. Adenosine, which is a neuromodulator for neurotransmitters through A1 adenosine receptors (A1AR), increases in vitro as a consequence of hyperammonemia [63,64]. Supposing that sleep-inducing effects of hyperammonemia are mediated by adenosine, Marini et al. reported that after sleep deprivation, hyperammonemic animals exhibited a larger increase in adenosine levels [63]. Moreover, functional imaging studies showed that patients with cirrhosis express lower A1AR binding by decreased density and affinity in cortical and subcortical regions compared to healthy controls, leading to further neurotransmitter imbalance and aggravation of HE [64].

The relation between sleep and thermoregulation has been extensively analyzed, both having a day-night rhythm strongly intercorrelated [65,66]. Normally, nocturnal secretion of melatonin determinates down regulation of core body temperature and increase of distal skin temperatures [65]. Skin temperature is elevated during night and low throughout the day, while core body temperature is on the contrary. Healthy people have the maximal potential to initiate and maintain sleep during the phase of lowered core body temperature [66]. Patients with CLD, cirrhosis in particular, have imbalanced body temperature regulation. In a case-control study, Garrido et al. investigated the skin temperatures and their gradient in relation with sleep-wake pattern among patients with cirrhosis [67]. The authors recorded the gradient between proximal and distal skin temperatures (DGP), which is considered a predictor of sleep latency. Sleep onset latency is dependent on peripheral vasodilation and redistribution of heat from core to peripherical sites [1]. It is well-known that cirrhotics with portal hypertension have a hyperdynamic circulatory syndrome with decreased systemic vascular resistance and splanchnic vasodilation [68]; therefore, they were proven to have higher proximal temperature in comparison than controls and they were unable to lose heat, recording high DGP values. These results correlated with a significantly longer sleep latency compared to patients without cirrhosis [67].

To summarize, the results presented so far regarding the circadian rhythm misalignment of patients with cirrhosis are controversial. On the one hand, melatonin clearance is the main mechanism incriminated. Melatonin impaired hepatic metabolism leads to abnormal patterns like high daytime plasma levels [30,51,52,53,54]. On the other hand, delays in the onset and acrophase of both melatonin and cortisol indicate, in addition, a central mechanism in the circadian disruption [49]. Other possible explanations consist of the presence of portal hypertension with portocaval shunts and the effects of ammonia and other neurotoxins upon SCN and its pathways [52,53,56]. Portal hypertension with hyperdynamic circulation is also responsible for impaired thermoregulation with high DGP. This was proven to be a predictor for the long sleep latency of cirrhotic patients [67,68].

### 4.3. SDs Attributed to the Etiology or Treatment of CLD

The occurrence of sleep disorders may also be attributed to the etiology or treatment of chronic liver disease:−NAFLD and OSA are epidemiologically linked and pathophysiologically overlapped [69]. OSA produces chronic intermittent hypoxia which promotes systemic inflammation, oxidative stress, insulin resistance, and adipose tissue dysfunction with serum lipid peroxidation. OSA is responsible, irrespective of other comorbidities, for NAFLD development and progression to cirrhosis [70]. Moreover, a strong association between severe OSA and NAFLD has been reported [71]. The bidirectional relation between OSA and liver is supported by high levels of HIF-1α (hypoxia-inducing factor), a serum protein and crucial transcription factor responsible for oxygen metabolism homeostasis [72]. Chronic intermittent hypoxia, seen in patients with OSA, increases the levels of HIF-1α in organs like brain and liver, aggravating the progression of NAFLD. HIF-1α promotes liver fibrosis in NAFLD by activating phosphatase and tensin homolog (PTEN)/p65 signaling pathway, which may be targeted for therapy [73]. The main symptom of OSA is excessive daytime sleepiness (EDS), being also the reason for which patients seek medical advice. The mechanisms implied in appearance of EDS are related to the consequences of OSA such as chronic sleep deprivation, intermittent hypoxia, and oxidative injuries in wake-promoting brain regions [74]. Gabryelska et al. suggested in a recent study that patients with OSA are also at risk for developing clock disruption, a process which might be mediated by HIF-1α, since its increased level was associated with the overexpression of circadian clock proteins [75]. Previously to these findings, a group of researchers investigated the expression level of mRNA coding for clock genes. They reported that this level was altered in OSA patients compared to healthy controls and did not decrease after one month of continuous positive airway pressure (CPAP) treatment [76]. Consistent with the aforementioned results, Yang et al. showed that the transcripts of all the investigated circadian clock genes displayed daily oscillation patterns in peripheral blood of controls, while three of them were arrhythmic in patients with OSA [77]. Studies also demonstrated dysregulation of 24-h melatonin secretion in patients with OSA, explained by correlations between OSA severity and low urinary 6-HMS, with elevated serum levels of melatonin in the afternoon [78,79].−Chronic hepatitis C. Sleep disorders were estimated in 65% of patients with chronic hepatitis C, independent of the antiviral treatment and before advanced stages of the disease [80]. Several researchers presented evidence which suggests that hepatitis C virus may cause brain dysfunction, even in the absence of severe hepatic disease or other risk factors [81,82,83]. It is unclear whether cerebral effects are connected to the pathogenesis of sleep complaints, but an association between sleep quality and immunological and virological biomarkers has been recorded [84,85]. The treatment of hepatitis C with interferon α (IFNα) brings additional risk of developing sleep symptoms, associated with depression. The underlying mechanisms may be related to the changes induced by IFN in serotonin metabolism and elevations in interleukin-6 (IL-6) and interleukin-1 (IL-1), known as sleep modulation cytokines [86].−Autoimmune cholestatic liver disease, primary biliary cirrhosis in particular, manifests with sleep symptoms (EDS, RLS) which are strongly associated with fatigue and pruritus [87,88]. Disorders like OSA and RLS are the main causes for EDS and fatigue encountered among these patients. EDS in these patients is neither correlated with the severity of the disease, nor with the presence of HE [87,89]. A possible incriminated mechanism is the elevated level of IL-6 which is responsible for mediating other sleep regulating cytokines (IL-1 and tumor necrosis factor—TNF) [90]. Furthermore, it is worth mentioning that administration of IL-1, TNF or IFNα in the cerebral ventricle of rabbits has been proven to induce NREM sleep. Increases in TNF levels are associated with shorter duration of sleep, while longer sleep time is associated with high levels of C-reactive protein and IL-6. Sochal et al. showed that sleep quality is affected in patients with inflammatory bowel disease, confirming that inflammation can lead to sleep disturbances which vice versa may affect the immune system [91].−Wilson’s disease and sleep disorders. The scarce publications reported a very wide frequency interval of SDs among these patients: 42–80% [92,93]. The main SDs are insomnia and RLS. Patients with Wilson’s disease complain of frequent nocturnal awakenings, sleep fragmentations, delayed wakeups in the morning, and EDS. These sleep abnormalities are caused by nocturia, associated psychiatric and behavioral comorbidities (such as depression and anxiety) or treatment with various drugs (such as dopaminergic therapy in high doses). Trindade et al. [94] reported another SD among patients with Wilson’s disease. RLS was present in 31% of the patients, in the absence of well-known associated factors (iron deficiency, neuropathy, chronic kidney disease) [94]. RLS in Wilson’s disease might be caused by accumulation of copper in thalamus, impairments in dopaminergic transmission and iron metabolism [95].−OSA was reported as a new and underdiagnosed complication of cirrhosis with ascites for the first time in 2003 by Crespo et al. [96]. After excluding subclinical HE, researchers conducted a prospective study which included 24 patients with alcohol- and viral-induced cirrhosis and ascites. The results showed that OSA could be a complication of high-volume ascites caused mainly by mechanical factors: diaphragmatic elevation led to decreased residual volume and obstruction of upper airways. Interestingly, the removal of ascitic fluid caused remission of OSA [96]. These findings are in line with those of Ogata et al. [97], who performed a larger study on 48 cirrhotic patients. They reported strong correlations between apnea-hypopnea index (AHI), as an objective measure of OSA, and volume of ascites. AHI was significantly higher in severe cirrhosis [97].

The literature also points out the fact that sleep disturbances in cirrhotics may be associated with impaired secretion of ghrelin [98,99]. Ghrelin is a hormone secreted by enteroendocrine cells, which plays an important role in food intake, sleep-wake cycle, and cognitive regulation, and may also be used as a marker to diagnose malnutrition in patients with CLD [98]. Normally, ghrelin stimulates appetite and triggers NREM sleep (slow-wave sleep), while sleep-deprivation increases the level of ghrelin. Bajaj et al. investigated the ghrelin secretion in cirrhotic patients with minimal HE. These patients exhibit lower SWS time (NREM sleep was absent in 80% of the patients), higher REM time, and significantly suppressed ghrelin levels, when they are compared to healthy controls [99].

## 5. Assessment of Sleep Disorders in Patients with CLD

### 5.1. Sleep Assessment

Regarding the tests used to assess SDs in patients with CLD, there are few and non-specific recommendations in the scientific literature. The manifestations of SDs (e.g., insomnia, EDS, OSA, RLS) are evaluated in terms of sleep quality and sleep-wake timing by means of various subjective and/or objective methods [1].

#### 5.1.1. Subjective Methods

These are based on daily sleep diaries and retrospective questionnaires.


*(a) Sleep Diaries*


In spite of the lack of specific recommendations or protocols to assess SD in CLD patients, a sleep diary is considered the “gold-standard” subjective method for assessing sleep, regardless of the background disease. Carney et al. [100] developed the “Consensus Sleep Diary” to evaluate sleep-wake disorders, in particular insomnia. Three versions of this sleep diary were created. The first one, the “Core Consensus Sleep Diary”, contains nine items (the time of getting into bed, the time of attempting to fall asleep, sleep onset latency, number of awakenings, duration of awakenings, time of final awakening, final rise time, sleep quality, and comments from respondent). The “Expanded Consensus Sleep Diary for Morning” (CSD-M) contains in addition items about early morning awakenings, napping, use of alcohol/caffeine/medication, which have indication to be completed in the morning. The third version, called the “Expanded Consensus Sleep Diary for Evening” (CSD-E) includes the same items as CSD-M, but they are grouped separately and the completion is done both in the morning and in the evening, respectively [98]. Several studies used sleep diaries to assess SDs among population with CLD [49,67,101,102].


*(b) The Pittsburgh Sleep Quality Index (PSQI)*


The Pittsburgh Sleep Quality Index (PSQI) is probably the most common used questionnaire to subjectively assess sleep quality. It was created to measure the sleep quality and patterns of sleep in adults over the preceding month. It consists of 19 items which are self-rated by the patient and are grouped into 7 components (subjective sleep quality, sleep latency, sleep duration, habitual sleep efficiency, sleep disturbances, use of sleeping medications, and daytime dysfunction), each having a score ranging between 0 and 3. The scores are summed, and the total score is ranging from 0 to 21, with the threshold of 5 points, dividing patients into “good sleepers” and “poor sleepers”. A total score of 5 or greater is indicative of a poor sleep, having a sensitivity of 89.6% and a specificity of 86.5% [103]. In a study conducted by Montagnese and colleagues [102], 70% of the cirrhotic patients were classified as “poor sleepers” according to PSQI.


*(c) Sleep Timing and Sleep Quality Screening Questionnaire*


In order to increase the awareness of SDs among patients with cirrhosis, a simplified version of PSQI was developed. The Sleep Timing and Sleep Quality Screening Questionnaire (STSQS) is rates sleep quality on an analogue scale (1 = the best sleep; 9 = the worst sleep ever) and sleep timing (bedtime, latency, arousals, wake-up, and get-up) [101]. STSQS takes only 1–2 min to complete, compared to PSQI which takes approximately 10 min to complete and 5 min to score. STSQS can be applied as a screening test to diagnose SDs as studies have reported significant correlations between the results obtained using STSQS and those obtained by PSQI [102,104]. In another study, the sensitivity of STSQS to discriminate cirrhotics with SDs from those without SDs was 79%, while the specificity was 65%, for a cut-off value of >5 [104].


*(d) The Epworth Sleepiness Scale (ESS)*


The Epworth Sleepiness Scale (ESS) is used to assess daytime somnolence or likelihood of falling asleep/“dozing off” in eight different situations of daily living on a Rensis Likert scale (0 = would never fall asleep in that situation, 1 = there is a slight chance, 2 = there is a medium chance, 3 = there is a high chance). The total score obtained by summing the score of each situation ranges from 0 to 24, while a score ≥11 is considered indicative of EDS [34]. Those eight daytime situations include: sitting and reading, watching TV, sitting inactive in a public place (e.g., theatre or meeting), sitting as a passenger in a car for an hour without a break, lying down to rest in the afternoon when circumstances permit, sitting and talking to someone, sitting quietly after a lunch without alcohol, and in a car while stopped for a few minutes in traffic [105]. The ESS was used as a subjective tool to assess diurnal sleepiness in patients with CLD in various studies, with comparable results [6,31,34,90,106,107]. Samanta et al. also showed in a cohort of 239 patients with cirrhosis that EDS correlates with cognitive impairment, the highest scores of ESS being recorded among patients with minimal HE compared with no minimal HE patients [34]. In accordance with the aforementioned study, other published data supported the association between EDS as indicated by ESS and HE [37,62]. However, these results were in contrast with those obtained by Montagnese et al., who did not find any association between EDS and HE per se but revealed significantly slower EEG in patients with EDS [2].


*(e) The Basic Nordic Sleep Questionnaire (BNSQ)*


The Basic Nordic Sleep Questionnaire (BNSQ) is a quantitative subjective tool which includes 27 different items in 21 standardized questions. This questionnaire assesses sleep complaints such as difficulty in initiating and maintaining sleep, EDS, night awakenings, early morning awakening, use of hypnotic drugs, napping habits, daytime sleepiness, snoring, and sleep apnea [108]. Three studies [31,109,110] investigated EDS by using BNSQ among liver transplantation recipients and patients with cirrhosis, reporting a high prevalence of EDS among these patients, with significant impact upon quality of life, even after liver transplantation [109].


*(f) STOP-Bang questionnaire*


The STOP-Bang questionnaire is a validated screening tool used to detect OSA among different patient populations [111]. It comes from the acronyms STOP (Snoring, Tiredness, Observed apnea and high blood Pressure) and Bang (Body mass index, Age, Neck circumference, Gender). It is easy and self-reportable and includes eight subjective and demographics items, each containing yes/no question. The total score ranges from 0 to 8 and a score ≥ 3 detects moderate to severe OSA with a sensitivity of 93–100% [111,112].


*(g) Berlin questionnaire (BQ)*


The Berlin questionnaire (BQ) is another validated method to assess the risk of OSA [113]. It has been used in 201 cirrhotic patients by Al-Jahdali and colleagues, who reported a risk of 42% for OSA as defined by BQ [36]. Singh et al. also revealed that BQ was indicative of OSA for almost half of the patients with NAFLD included in their study [114].


*(h) The International Restless Leg Syndrome Study Group rating scale (IRLSS)*


The International Restless Leg Syndrome Study Group rating scale (IRLSS) is a validated instrument to assess RLS presence and severity. IRLSS criteria have been fulfilled in 29% of patients with primary biliary cirrhosis, according to Anderson et al. [89].

#### 5.1.2. Objective Methods

They are known also as polysomnography and home sleep apnea testing and allow more precise assessment but are used as second-line investigations and only in 10–25% of patients with sleep-wake disorders. Several studies have used objective tools like actigraphy or polysomnography to investigate SDs diagnoses in patients with CLD [27,99,101,115,116,117,118,119,120,121].

(a)Polysomnography (PSG) encompasses electroencephalogram, electrooculogram, electromyogram, and measurements of nasal and oral airflow. PSG is currently the “gold standard” diagnostic test for OSA and other sleep disorders, but may have limited access, being an expensive, time-consuming, in-laboratory test [1,38,122]. In 1972, Kurtz et al. opened the area of electroencephalography and neurophysiology in cirrhosis by investigating the EEG recordings of patients with different stages of encephalopathy [123]. Later, Teodoro and colleagues [115] showed that patients with cirrhosis experience an increased REM latency and reduced REM sleep. Recently, a group of researchers [117] assessed the prevalence of sleep-disordered breathing, OSA in particular, among cirrhotic patients of viral etiology. The evaluation was done through subjective tools and a full-night PSG sleep study. It resulted that cirrhotics had a significantly higher percentage of OSA (56.2%) compared to healthy controls (12.5%), while there were no significant differences between groups in terms of sleep efficiency and base SpO_2_ [117].(b)Home sleep apnea test (HSAT), also known as unattended sleep testing or portable monitoring, is an alternative less expensive and more convenient than polysomnography but has some disadvantages: it does not typically include electroencephalography, electrooculography, or electromyography sensors and might underestimate the severity of OSA [124]. Portable devices should measure peripheral arterial tonometry (PAT), oximetry, heart rate, snoring, wrist activity (actigraphy), and body position section [125].(c)Actigraphy is considered a semi-quantitative method which records the patient’s locomotor activity by means of an accelerometer. The recorded data are further analyzed via a software which estimates sleep parameters [1]. Actigraphs have been used together with sleep diaries and questionnaires in several studies in order to assess the prevalence of SDs in patients with CLD before and after therapeutic management [119,121,126].

Most data published regarding SDs in patients with CLD were obtained by self-reported questionnaires, sleep diaries, actigraphy, and some were assessed through polysomnography (Table 3).

### 5.2. Neuropsychiatric Assessment

Depending on severity, HE may be divided in covert HE and overt HE. Covert HE includes minimal HE (MHE) and grade I HE according to West Haven Criteria [127]. Overt HE is clinically diagnosed, being a complication of portal hypertension and decompensated cirrhosis. The diagnosis of covert HE needs to be made by psychometric tests, since the clinical examination is normal. The psychometric hepatic encephalopathy score (PHES) is a standardized method to assess neuropsychological impairments of MHE and is made of five tests: number connection test-A, number connection test-B, serial dotting test, line tracing test, and digit symbol test [128]. Other tests described to assess covert HE include electroencephalography, sensory and cognitive evoked potentials, and neuroimaging [5,129].

In practice, the prevalence of MHE varies between 30% and 70% and is suspected in cirrhotic patients who have abnormalities on psychometric testing, particularly impaired attention and reaction speed [130]. It has also been demonstrated that PHES score is significantly correlated with PSQI and ESS scores in a study conducted among 100 patients with cirrhosis [34]. Thus, poor sleep parameters are associated with impaired cognition [34]. Unlike these results, Cordoba et al. [27] and later Bianchi and coworkers [28] showed there was no difference regarding prevalence of SDs in relation with PHES.

Conventional psychometric tests are time consuming and usually require a trained specialist. Therefore, free downloadable applications have been developed. An example is Stroop test, an innovative computerized test, which was designed by Bajaj et al. based on the paper-and-pencil [131]. It has high sensitivity and specificity and can be used to rapidly diagnose MHE in clinical practice, having been already validated in multicenter studies [132].

Another tool used in research for investigating SDs and HE is actigraphy. Cordoba and his colleagues showed that actigraphy can diagnose decreased motor activity in patients with cirrhosis without evidence of manifesting HE [27]. Later, researchers from France [133] investigated sleep-wake patterns in HE by using actigraphy as a diagnostic tool and concluded that motor activity correlated with HE clinical stages.

### 5.3. Health-Related Quality of Life (HRQOL)

Health-related quality of life (HRQOL) is quantified through short form (SF)-36 questionnaire, which contains 36 multiple-choice questions regarding physical and mental health [134]. Samanta et al. [34] revealed in their study that night-time sleep disorders assessed by PSQI are associated with impaired HRQOL. Moreover, Montagnese and colleagues [2] published a research study showing that poor HRQOL was not only associated with night-time SDs, but also EDS and evening preference. A recent study including liver transplant patients showed via PSQI questionnaire that the high prevalence and incidence of poor sleep quality among these patients is significantly correlated with depression, anxiety, and social support [135]. Moreover, it has been postulated that cognitive deficits seen in cirrhotics are interlinked with SDs, both having negative consequences on quality of life [136].

HRQOL can also be assessed by the Chronic Liver Disease Questionnaire (CLDQ), which is a liver-disease specific instrument that includes 29 items grouped in 6 domains: abdominal symptoms, systemic symptoms, activity, emotional function, fatigue, and worry [137]. In a study [29] which enrolled 193 subjects diagnosed with cirrhosis, the authors assessed sleep quality through PSQI and actigraphy. The results showed that disturbed sleep is an independent predictor for decreased quality of life [29].

## 6. Management of Sleep Disorders in Patients with Chronic Liver Disease

Given that poor sleep quality is connected with manifestations of cognitive dysfunction in patients with cirrhosis, treatment strategies should focus both on sleep and cognition [136].

### 6.1. Therapeutic Options for HE

Cognitive impairment is a sign of overt HE but may also be present in covert HE. Even in the mildest stage, HE has been proven to influence quality of life and sleep-wake patterns, with clinical significance predisposing to development of overt HE and increased mortality. MHE should be early detected and treated [138]. The pharmacological treatments for HE include non-absorbable disaccharides (lactulose), antibiotics (rifaximin), oral branched-chain amino acids (BCAAs), intravenous L-ornithine L-aspartate (LOLA), probiotics [139]. Studies [118,139] have shown that lactulose administration in patients with cirrhosis and MHE significantly improves total sleep time, sleep parameters and overall efficiency assessed by subjective tools and PSG. In addition, after threemonths treatment with lactulose, researchers observed major improvements in cognitive function and PHES score and also in HRQOL parameters [118,140]. Bruyneel et al. [119] also investigated the impact of rifaximin treatment in recurrent HE referring to sleep-wake parameters measured by actigraphic recordings and unattended 24-h PSG. The results revealed that the level of motor activity, REM sleep length, and sleep efficiency were decreased before rifaximin. After 28 days of rifaximin treatment, sleep architecture improved by increase in REM sleep. However, no improvement was reported in HRQOL and physical activity [119].

For the first time, Spahr and his colleagues [121], evaluated the effects of an antihistamine drug (histamine H1 blocker hydroxyzine) on sleep alterations in patients with cirrhosis and MHE. They reported a significant subjective improvement in sleep in 40% of the patients and a ≥30% increase in sleep efficiency (measured by actigraphy) in 65% of the patients. The recommendations of this drug use are limited and related to the risk of precipitating overt HE [122]. Bhat et al. showed that liver transplant significantly improved sleep parameters in patients with alcoholic liver disease compared with hepatitis C patients [109].

### 6.2. Therapeutic Options for Sleep Disorders

According to various guidelines, sleep disorders may benefit from both pharmacological and non-pharmacological therapies [141,142]. The majority of studies assess these therapies for primary sleep disorders. Still, when we recommend them in patients with CLD it is necessary to emphasize that most drugs are contraindicated due to severe adverse reactions. Practically, benzodiazepines, benzodiazepine receptor agonists, antidepressants, antipsychotics, antihistamines, which have been studied in most SDs, are well-known for their risk for addiction, tolerance, and cognition impairment [143].

There are limited data about possible therapeutic options in patients with CLD.

Modafinil, a potent suppressor in hepatocytes of CYP2C9, approved by FDA in treatment of narcolepsy [144], has been studied as a treatment option to overcome EDS and fatigue in patients with primary biliary cirrhosis [145,146]. Of the patients enrolled in an open-labeled study, 66% could not tolerate long-term treatment, but 86% of the remaining patients had an effective response by properly controlling EDS and having normal ESS [146]. In a double-blind study, modafinil also improved subjective daytime sleepiness in patients with OSA and regular CPAP use, but who still complained of EDS [147]. Further placebo-controlled trials need to be conducted in order to validate this treatment in current practice of SDs in CLD.

Zolpidem was another therapeutic option which was assessed for its effectiveness among cirrhotic patients [148]. Zolpidem 5 mg for 4 weeks proved to be safe, effective and well-tolerated among a population of patients with Child–Turcotte–Pugh class A and B cirrhosis. The authors of the study reported significative improvements in total sleep time, efficiency, and polysomnographic parameters [148].

The protective effects of melatonin in liver diseases have been studied in several studies. Most data come from those where melatonin supplements were assessed as a pharmacological treatment for NAFLD. Due to its antioxidant action, melatonin has beneficial effects such as decreasing levels of liver enzymes and improving liver histology [149,150]. Moreover, in association with tryptophan, melatonin reduces levels of IL-1, IL-6, and TNFα in patients with NAFLD [151]. Although melatonin supplementation has not been specifically evaluated in CLD, it was proven to play an important role in primary sleep disorders, by increasing total sleep time, decreasing sleep onset latency, and improving overall sleep quality [152]. Taking into consideration that melatonin treatment has no visible short or long-term adverse effects, it might be an option for SDs among patients with CLD, therefore making further investigation necessary.

Continuous positive airway pressure (CPAP) is considered the treatment of choice for OSA and is also indicated for improving self-reported sleepiness [153]. Oxidative stress induced by nocturnal intermittent hypoxia is reversible with CPAP therapy, leading to a significant reduction in leptin level (known as satiety hormone) and insulin resistance, respectively [154]. Observational studies suggest that CPAP might also improve daytime symptoms, quality of life, and the evolution of NAFLD [155], by decreasing liver enzymes [156,157].

In patients with hepatitis C virus (HCV) infection, and treatment with Peg-IFN-α and Ribavirin, supplementation with resveratrol led to lower scores of sleep questionnaires (PSQI and ESS) and reduction in anxiety, depression, and SDs symptoms [158]. Moreover, there were studies showing that the type of antiviral therapies used in chronic hepatitis C might influence the prevalence of SDs [116,159]. A recent study [159] compared the severity of SDs detected by actigraphy and PSQI between patients treated with interferon-based therapy and those who were administered interferon-free direct-acting antiviral regimens (DAA). The authors state that DAA are less prone to causing sleep problems, suggesting that depression and low sleep quality can be remedied after switching interferon-based HCV treatment with DAAs, which is similar to other published data [159,160,161].

Non-pharmacological treatments for SDs have also been evaluated over the last years, with limited beneficial results. De Rui et al. [126] studied the effects of bright daytime blue-enriched light on the circadian rhythm of patients with decompensated cirrhosis, but the results showed no improvements of sleep parameters. Unlike these results, the response to light studied among patients with primary biliary cholangitis was positive, in terms of sleep time and quality [162]. The subjects were exposed for 45 min to bright light, immediately after getting up. As a result, the participants experienced less daytime sleepiness, improved nighttime sleep quality, and earlier sleep on-set. There was no significant change regarding health-related quality of life after exposure to light [162]. Exposure to bright light in the early morning and avoidance in the evening might be effective in cirrhotic patients, but still has to be further investigated.

Other approaches, including mindfulness-based stress reduction, neuromuscular relaxation training, and lavender warm baths improve sleep-related symptoms in patients with CLD [163].

## 7. Conclusions

This review addresses an important area regarding sleep disorders and their association with CLD. The nature of SDs in patients with CLD remains poorly understood. Given that the published data in literature are limited, future prospective studies are necessary to clarify the pathogenesis of each sleep disorder in accordance with the etiology of liver disease. Hence, success in assessing and treating SDs encountered in CLD might be achieved by a more individualized strategy. Nevertheless, the evolution and quality of life would also be improved, as it is well-known that SDs represent a major distress for patients.

## Figures and Tables

**Figure 1 brainsci-11-00142-f001:**
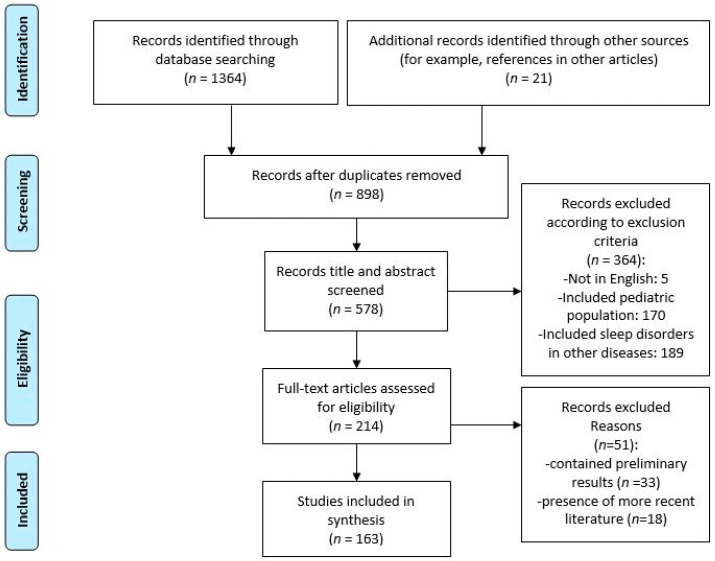
Flow chart used to perform the review.

**Figure 2 brainsci-11-00142-f002:**
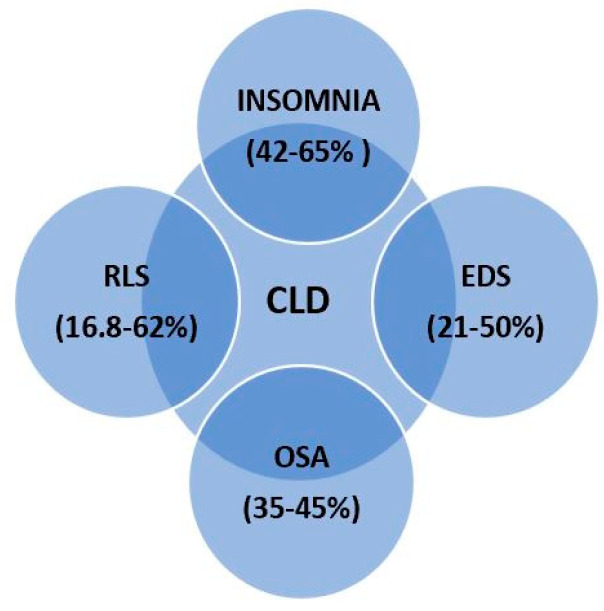
The most frequent phenotypes of SDs in chronic liver disease (CLD) and their prevalence. Legend: EDS—excessive daytime sleepiness; OSA—obstructive sleep apnea; RLS—restless leg syndrome.

**Figure 3 brainsci-11-00142-f003:**
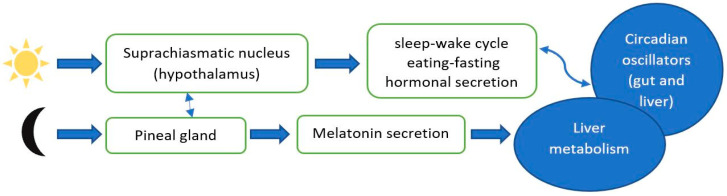
Schematic illustration of circadian rhythm.

**Table 1 brainsci-11-00142-t001:** Classification of sleep disorders (SDs) according to the International Classification of Sleep Disorders (ICSD) 3rd edition [8].

Groups of SDs	Classification of Main SDs	References
**Insomnia**	Chronic insomnia disorder Short-term insomnia disorder Other insomnia disorders	Sateia MJ [9] Morin et al. [10]
**Sleep-related breathing disorders**	OSA disorders CSA syndromes Sleep-related hypoventilation disorders Sleep-related hypoxemia disorder	Sateia MJ [9] Foldvary-Schaefer et al. [11] Mohammadieh et al. [12]
**Circadian rhythm sleep-wake disorders**	Sleep-wake phase disorders Sleep-wake rhythm disorders Non-24-h sleep-wake rhythm disorder Shift work disorder Jet lag disorder Circadian sleep-wake disorder not otherwise specified	Sateia MJ [8] Culnan et al. [13] Spiegelhalder et al. [14]
**Parasomnias**	NREM-related parasomnias REM-related parasomnias Other parasomnias	Sateia MJ [8] Bollu et al. [15]
**Sleep-related movement disorders**	Restless legs syndrome Sleep-related bruxism Sleep-related movement disorder due to a medical disorder Sleep-related movement disorder due to a medication or substance Sleep-related movement disorder, unspecified	Sateia MJ [8] Trotti et al. [16]
**Central disorders of hypersomnolence**	Narcolepsy Hypersomnia due to a medical disorder Hypersomnia due to a medication or substance Hypersomnia associated with a psychiatric disorder Insufficient sleep syndrome	Khan et al. [17] Dauvilliers Y et al. [18]
**Other sleep disorders**	SDs that cannot be classified elsewhere in the ICSD: (a) Sleep-related medical and neurological disorders (b) Substance-induced sleep disorders	Rains JC [19]

Legend: OSA—obstructive sleep apnea, CSA—central sleep apnea, NREM—non-rapid eye movement sleep, REM—rapid eye movement sleep.

**Table 2 brainsci-11-00142-t002:** The etiology of chronic liver disease [4,46].

Etiology of CLD
Alcohol-related liver disease Viral hepatitis B Viral hepatitis C Non-alcoholic fatty liver disease (NAFLD) Autoimmune liver disease (primary biliary cholangitis, autoimmune hepatitis, primary sclerosing cholangitis) Genetic conditions (Wilson disease, hemochromatosis, alpha-1-antitrypsin deficiency)

**Table 3 brainsci-11-00142-t003:** Assessment of sleep disorders in CLD.

Methods for Assessing SDs	Tools Used in Scientific Literature to Assess SDs in CLD	References
**Subjective**	Sleep diaries	[49,67,101,102,120]
	The Pittsburgh Sleep Quality Index (PSQI)	[102,103,116,118,119]
	Sleep Timing and Sleep Quality Screening Questionnaire (STSQS)	[102,103,104]
	The Epworth Sleepiness Scale (ESS)	[6,31,34,37,62,90,105,106,107,118]
	The Basic Nordic Sleep Questionnaire (BNSQ)	[31,108,109,110]
	The Horne–Ostberg (HO) questionnaire	[27,110]
	STOP-Bang questionnaire	[111,112]
	Berlin questionnaire (BQ)	[36,114]
	The International Restless Leg Syndrome Study Group rating scale (IRLSS)	[89]
**Objective**	Polysomnography	[99,115,117,118,119,123]
	Actigraphy	[27,102,116,119,120,121,126]

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
