# Peer review of "Comprehensive Overview of Sleep Disorders in Patients with Chronic Liver Disease"

_brainsci, 2021, doi:10.3390/brainsci11020142_

Round 1

Reviewer 1 Report

Manuscript ID: brainsci-1054771-peer-review-v1

Title: Comprehensive overview of sleep disorders in patients with chronic liver disease

Journal: Brain Sciences

Abstract

  1. Authors should describe the main results of their review.

Introduction

  1. Page 2, line 57. Authors should replace “data basis” with “databases”.
  2. Page 2, line 58. Elsevier, MDPI, and Springer are publishers. Authors should look for papers only in citation databases.
  3. Page 2, lines 58-59. Authors wrote that “more than 150 references being selected to reference the statements in this paper”. How did authors select these articles?
  4. Authors should at minimum perform a systematic review following the PRISMA guidelines (https://journals.plos.org/plosmedicine/article?id=10.1371/journal.pmed.1000097). Moreover, for example, they should specify the inclusion/exclusion criteria and the key words used for the bibliographic search.

Epidemiology and phenotypes of SDs in CLD

  1. From page 4, line 130 to page 5, line 137. The reviewed studies reported extremely different prevalence. Which can be a possible explanation of such a great discrepancy between studies?

Discussion

  1. Authors should avoid describing sleep. In the discussion section, they are expected to discuss the results of the studies being reviewed.
  2. Page 6, lines 163-164. Authors wrote “During REM, the mental activity is associated with dreaming”. Dreaming is associated also with slow wave sleep (https://academic.oup.com/sleep/article/15/6/562/2749336?login=true).
  3. Page 6, line 203. Authors should replace “acro phase” with “acrophase”.
  4. Page 7, lines 210-211. Authors wrote “melatonin profile abnormalities do not explain the sleep –wake disturbances exhibited by these patients”. Since this is a direct quotation, authors should add the number of page of the original study from which this sentence has been extracted.

General comment on assessment of sleep disorders in patients with CLD and management of sleep disorders in patients with chronic liver disease

  1. These sections should be part of a different study.

Assessment of sleep disorders in patients with CLD

  1. Page 9, line 319. Authors wrote “In spite of the lack of standardization”. Actually, a standardized version of the sleep diary has been proposed by Carney and colleagues (2012) in a paper that has been quoted by authors one line below.
  2. Page 9, line 333. Authors wrote “PSQI is probably the most common used test to subjectively assess SDs.”. PSQI is a questionnaire and not a test. Moreover, PSQI does allow to measure the perceived sleep quality and not to assess sleep disorders.
  3. Page 10, line 363. ESS allows to measure diurnal sleepiness and not sleep disorders.
  4. Page 10, line 381. The Morningness-Eveningness Questionnaire does not allow to measure sleep. Such questionnaire allows to measure one of the most marked interindividual differences in circadian rhythms, i.e., the morningness-eveningness preference.
  5. Page 11, line 420. Authors should replace “efficacy” with “efficiency”.
  6. Page 11, line 427. Actigraphy should be presented in a separate paragraph.

Author Response

Journal: Brain Sciences

Manuscript: brainsci-1054771

Dear Reviewer 1

We are very thankful to the Reviewers for their notes, time and support in improving our paper; we have carefully read the comments and have revised / completed the manuscript accordingly. Our responses are given in a point-by-point manner below (in blue), as well all the changes to the manuscript are highlighted in red.

In order to improve the quality of the manuscript, the text was modified, completed, and corrected, respecting exactly the recommendations of the 2 reviewers.  We hope that in the revised form, our manuscript will be found as suitable for publication.

Abstract 

1. Authors should describe the main results of their review.

 Response: Thank you very much for your suggestion. We limited our abstract to less than 200 words as required and considered to reorganize the text in a short introduction about the importance of the topic, followed by the aim of this paper. Afterwards, we described the main results, as you suggested.

Page 1, lines 20-33: “The impact of sleep disorders (SDs) on patients with chronic liver diseases (CLD) is tremendous. SDs are frequently encountered among these patients and interfere with their quality of life. This review aims to present the data available so far about the prevalence, phenotypes and proposed pathophysiological mechanisms of SDs in CLD. Moreover, we proposed to search the literature regarding the most reliable methods to assess SDs and the possible therapeutic options in patients with CLD. The main results of this review show that when it comes to prevalence, the percentages reported vary widely between studies performed among populations from the USA or Europe and those coming from Asian countries. Furthermore, it has been proved that SDs may also be present in the absence of neurocognitive disorders attributable to hepatic encephalopathy (HE), which contradicts traditional suppositions where SDs were considered part of the clinical scenario of HE. Currently, there are no specific recommendations or protocols to assess SDs in CLD patients and data about the therapeutic management are limited. Taking into consideration their impact, a protocol for diagnosing and managing SDs should be developed and included in the daily practice of hepatologists.“

Introduction

1. Page 2, line 57. Authors should replace “data basis” with “databases”.

Response: We have replaced “data basis” with “databases” (page 2, line 57), as suggested.

2. Page 2, line 58. Elsevier, MDPI, and Springer are publishers. Authors should look for papers only in citation databases.

Response: Thank you for your observation. We corrected and specified the databases that were searched for this review.

Page 2, lines 57-58: “PubMed Central, Cochrane Library, EMBASE”

3. Page 2, lines 58-59. Authors wrote that “more than 150 references being selected to reference the statements in this paper”. How did authors select these articles?

Response: We have added the following paragraph to clarify the selection: (lines 57-64)

Page 2, lines 57-64: “Electronic databases (PubMed Central, Cochrane Library, EMBASE) were searched, using the following keywords: “sleep disorders”, “sleep disturbances”, “chronic liver disease”, “cirrhosis”, “hepatitis”, “nonalcoholic fatty liver disease”. We included in this systematic review 161 publications following PRISMA guidelines. The inclusion criteria were human and animal studies where sleep disorders and liver disease coexisted. All relevant articles in English language or with English abstracts were retrieved. In addition, cross-references from the identified articles were reviewed. Studies referred to pediatric population or sleep disorders encountered in other diseases were excluded, in the absence of hepatic impairment.”

4. Authors should at minimum perform a systematic review following the PRISMA guidelines (https://journals.plos.org/plosmedicine/article?id=10.1371/journal.pmed.1000097). Moreover, for example, they should specify the inclusion/exclusion criteria and the key words used for the bibliographic search.

 Response: We wrote our review after consulting PRISMA checklist and according to its guidelines; we specified key words, inclusion and exclusion criteria. (page 2, lines 57-64)

Epidemiology and phenotypes of SDs in CLD

1. From page 4, line 130 to page 5, line 137. The reviewed studies reported extremely different prevalence. Which can be a possible explanation of such a great discrepancy between studies?

Response: Thank you for your observation. We added the following paragraph:

Page 4, lines 142-146: “The possible explanation for the great discrepancy between studies comes from the differences in the populations of patients with chronic liver disease (CLD) and restless legs syndrome (RLS). A higher prevalence is seen in studies performed in the USA or Europe compared to Asia. Studies have shown that racial differences exist not only for idiopathic RLS, but alsosecondary RLS [43].”

Discussion

1. Authors should avoid describing sleep. In the discussion section, they are expected to discuss the results of the studies being reviewed.

 Response: Thank you very much for your suggestions. We removed the definition of sleep from this subchapter. We considered important to maintain several basic explanations regarding the normal pathophysiology of sleep, including its two stages with their features, as being useful for clinicians, in particular, to understand the abnormalities that appear in patients with CLD and to integrate their mechanisms in the overall body functioning. We also added the following paragraphs in completion to the results of the studies discussed in this chapter:

Page 7, lines 267-275: “To summarize, the results presented so far regarding the circadian rhythm misalignment of patients with cirrhosis are controversial. On the one hand, melatonin clearance is the main mechanism incriminated. Melatonin impaired hepatic metabolism leads to abnormal patterns like high daytime plasma levels [30, 51-54]. On the other hand, delays in the onset and acrophase of both melatonin and cortisol indicate, in addition, a central mechanism in the circadian disruption [49]. Other possible explanations consist in the presence of portal hypertension with portocaval shunts and the effects of ammonia and other neurotoxins upon SCN and its pathways [52, 53, 56]. Portal hypertension with hyperdynamic circulation is also responsible for impaired thermoregulation with high DGP. This was proved to be a predictor for the long sleep latency of cirrhotic patients [67, 68].”

Page 7, lines 283-289: “The bidirectional relation between OSA and liver is supported by high levels of HIF-1α (hypoxia-inducing factor), a serum protein and crucial transcription factor responsible for oxygen metabolism homeostasis [72]. Chronic intermittent hypoxia, seen in patients with OSA, increases the levels of HIF-1α in organs like brain and liver, aggravating the progression of NAFLD. HIF-1αpromotes liver fibrosis in NAFLD by activating PTEN/p65 signalling pathway, which may be targeted for therapy [73].“

Page 8, lines 292-296: “The circadian clock misalignment is also encountered in OSA as a consequence of chronic sleep deprivation and hypoxia-reoxygenation, which have adverse effects on wake-promoting neurons in basal forebrain and brainstem. Besides, it has been shown that structural brain changes in white and gray matter are present in patients with OSA, leading to cognitive deficits and sleepiness [74].“

Page 8, lines 314-319: “Furthermore, it is worth mentioning that administration of IL-1, TNF or IFNα in the cerebral ventricle of rabbits has been proven to induce NREM sleep. Increases in TNF levels are associated with shorter duration of sleep, while longer sleep time is associated with high levels of C-reactive protein and IL-6. Sochal et al showed that sleep quality is affected in patients with inflammatory bowel disease, confirming that inflammation can lead to sleep disturbances which vice versa may affect the immune system [88].”

2. Page 6, lines 163-164. Authors wrote “During REM, the mental activity is associated with dreaming”. Dreaming is associated also with slow wave sleep (https://academic.oup.com/sleep/article/15/6/562/2749336?login=true).

Response: Thank you for mentioning that. We searched for the article that you provided and indeed authors report that dreaming is also associated with SWS as you have stated; therefore, we decided that in order to avoid misconception it is better to remove this information from our paper, as we considered it redundant for the topic.

3. Page 6, line 203. Authors should replace “acro phase” with “acrophase”.

 Response: We replaced “acro phase” with “acrophase” (page 6, lines 216 and 218) as suggested.

4. Page 7, lines 210-211. Authors wrote “melatonin profile abnormalities do not explain the sleep –wake disturbances exhibited by these patients”. Since this is a direct quotation, authors should add the number of page of the original study from which this sentence has been extracted.

 Response: Thank you for your observation. Accordingly, we added the author, year and page of the original study from which we took the quotation.

Page 6, lines 222-224: “In an observational study, the authors concluded that “melatonin profile abnormalities do not explain the sleep –wake disturbances exhibited by these patients” (Montagnese et al, 2010, p. 1781);“

General comment on assessment of sleep disorders in patients with CLD and management of sleep disorders in patients with chronic liver disease

1. These sections should be part of a different study.

 Response: Thank you very much for your suggestion. We included the sections “assessment and management of sleep disorders in patients with CLD” because we wanted to create a comprehensive review. Sleep disorders have a tremendous impact on this population of patients and the complete information we presented here has the purpose to increase awareness among doctors.

Assessment of sleep disorders in patients with CLD

1. Page 9, line 319. Authors wrote “In spite of the lack of standardization”. Actually, a standardized version of the sleep diary has been proposed by Carney and colleagues (2012) in a paper that has been quoted by authors one line below.

 Response: Thank you for your observation. We changed the expression as it was misleading with a new one:

Page 9, lines 356-358: “In spite of the lack of specific recommendations or protocols to assess SD in CLD patients, sleep diary is considered the “gold-standard” subjective method for assessing sleep, regardless of the background disease.”

2. Page 9, line 333. Authors wrote “PSQI is probably the most common used test to subjectively assess SDs.”. PSQI is a questionnaire and not a test. Moreover, PSQI does allow to measure the perceived sleep quality and not to assess sleep disorders.

 Response: We changed the word “test” with “questionnaire”(page 9, line 370) to be accurate, as it is a subjective method, not a test (as you suggested). We included PSQI in the subjective assessment because is one of the most important questionnaires used among studies in this population to assess the quality of sleep and to correlate its score or its parameters with results obtained from other questionnaires or tests which may diagnose sleep disorders.

3. Page 10, line 363. ESS allows to measure diurnal sleepiness and not sleep disorders.

 Response: Thank you for mentioning it. Consequently, we changed the expression as follows:

Page 9, lines 398-399: “The ESS was used as a subjective tool to assess diurnal sleepiness in patients with CLD in various studies, with comparable results [6, 31, 34, 87, 103, 104].”

4. Page 10, line 381. The Morningness-Eveningness Questionnaire does not allow to measure sleep. Such questionnaire allows to measure one of the most marked interindividual differences in circadian rhythms, i.e., the morningness-eveningness preference.

 Response: We included this questionnaire because it was used by researchers as a tool to determine chronotypology in cirrhosis and correlate it to melatonin levels which are unbalanced in these patients. (https://doi.org/10.1186/1740-3391-7-6)

5. Page 11, line 420. Authors should replace “efficacy” with “efficiency”.

 Response: We replaced “efficacy” with “efficiency” (page 11, line 449), as suggested.

6. Page 11, line 427. Actigraphy should be presented in a separate paragraph.

Response: Thank you for your observation. We presented Actigraphy in a separate paragraph (page 11, lines 456-460), as suggested.

With all our gratitude for your help and support,

The Authors

Reviewer 2 Report

Manuscript entitled „Comprehensive overview of sleep disorders in patients with chronic liver disease” summarizes available data on sleep disorders in patients with chronic liver diseases.

Table 1 is unnecessary for the article; it does not need the full table of sleep disorders qualifications. The summary of main groups and short description of each symptom or disorder when first discussed should be sufficient.

The heading of 4thchapter as discussion is misleading. First part could be named “circadian clock misalignment. Additionally, while further discussing OSA in context of CLS it could be interesting to discuss data showing the circadian misalignment in OSA itself.

The chapter regarding tools used in the SD assessment would more fit earlier in the manuscript, possibly after “Classification of SDs”. Furthermore, it should be shortened as it summarizes general available tools, they are not specific for CLC.

I would advise to expand on possible mechanisms of SD in CLC as it is well described with circadian misalignment as possible cause. Here particularly consider possible effect of drugs used in the treatment such as glucocorticoids (fe. doi: 10.1016/j.smrv.2020.101380) or possiblt anti-TNF (fe. doi: 10.3390/jcm9092921) as well as the other medications on sleep.

Additionally, it would be very interesting to mention and discuss possible two-sided relationship between sleep disorders and liver diseases. Especially obstructive sleep apnea might be a cause of liver fibrosis due to chronic effects of hypoxia. Possible pathways should be presented such as through HIF-1alpha (fe. doi: 10.5664/jcsm.8682).

Author Response

Journal: Brain Sciences

Manuscript: brainsci-1054771

Dear Reviewer 2

We are very thankful to the Reviewers for their notes, time and support in improving our paper; we have carefully read the comments and have revised / completed the manuscript accordingly. Our responses are given in a point-by-point manner below (in blue), as well all the changes to the manuscript are highlighted in red.

In order to improve the quality of the manuscript, the text was modified, completed, and corrected, respecting exactly the recommendations of the 2 reviewers.  We hope that in the revised form, our manuscript will be found as suitable for publication.

Manuscript entitled „Comprehensive overview of sleep disorders in patients with chronic liver disease” summarizes available data on sleep disorders in patients with chronic liver diseases.

Table 1 is unnecessary for the article; it does not need the full table of sleep disorders qualifications. The summary of main groups and short description of each symptom or disorder when first discussed should be sufficient.

Response: Thank you very much for your comment. We simplified Table 1 (page 2, line 73) and constrained the information to the large groups and some of the main sleep disorders. We thought we should keep the table (simplified) because we consider essential to include the classification of these disturbances in a review about sleep disorders.

The heading of 4th chapter as discussion is misleading. First part could be named “circadian clock misalignment. Additionally, while further discussing OSA in context of CLS it could be interesting to discuss data showing the circadian misalignment in OSA itself.

Response: Thank you for your suggestion.  According to it, we divided the 4thchapter in subchapters: 4.1. Etiology of CLD (page 4, line 152); 4.2. Circadian clock misalignment in CLD (page 5, line 162); 4.3. SDs attributed to the etiology or treatment of CLD (page 7, line 276).

The subject referring to circadian misalignment in OSA is very interesting. We reported further from line 292 to 296 (page 7) the main results explaining this finding, which of course should be extensively reviewed in future papers. The following text was added:

“The circadian clock misalignment is also encountered in OSA as a consequence of chronic sleep deprivation and hypoxia-reoxygenation, which have adverse effects on wake-promoting neurons in basal forebrain and brainstem. Besides, it has been shown that structural brain changes in white and gray matter are present in patients with OSA, leading to cognitive deficits and sleepiness [74].”

Then, we continued with the modifications observed in melatonin secretion in patients with OSA.

The chapter regarding tools used in the SD assessment would more fit earlier in the manuscript, possibly after “Classification of SDs”. Furthermore, it should be shortened as it summarizes general available tools, they are not specific for CLC.

Response: Thank you for your suggestion. We agree with your observation, but the reason for which we preferred to write this chapter later in the review is related to the fact that a previous description of SD phenotypes and their mechanisms in CLD leads to a better understanding of the tools described (as they were used by researchers in patients with CLD).

Secondly, we considered the assessment of SD an important chapter for this review and treated it extensively, as we wished to create a comprehensive review.

I would advise to expand on possible mechanisms of SD in CLC as it is well described with circadian misalignment as possible cause. Here particularly consider possible effect of drugs used in the treatment such as glucocorticoids (fe. doi: 10.1016/j.smrv.2020.101380) or possiblt anti-TNF (fe. doi: 10.3390/jcm9092921) as well as the other medications on sleep.

Response: Thank you very much for your advice. We searched for what you proposed and expanded on possible mechanisms accordingly, by inserting 2 more paragraphs and the 2 references that you suggested:

Pages 5-6, lines 188 – 194:”The 24h sleep-wake cycle is influenced by glucocorticoid hormones, known as essential messengers. Their action upon SCN is dual, involving serotonin depletion and reducing arginine vasopressin signaling in the SCN. Glucocorticoid treatment administered in numerous inflammatory diseases may cause sleep architecture abnormalities (increased REM latency, increased time spent awake and SWS). Similarly, to glucocorticoids, statins may lead to disruption in the SCN signaling via metabolic impairments, such as release of muscle metabolites, namely prostaglandins [50].”

Page 8, lines 314 – 319: “Furthermore, it is worth mentioning that administration of IL-1, TNF or IFNα in the cerebral ventricle of rabbits has been proven to induce NREM sleep. Increases in TNF levels are associated with shorter duration of sleep, while longer sleep time is associated with high levels of C-reactive protein and IL-6. Sochal et al showed that sleep quality is affected in patients with inflammatory bowel disease, confirming that inflammation can lead to sleep disturbances which vice versa may affect the immune system [88].”

Additionally, it would be very interesting to mention and discuss possible two-sided relationship between sleep disorders and liver diseases. Especially obstructive sleep apnea might be a cause of liver fibrosis due to chronic effects of hypoxia. Possible pathways should be presented such as through HIF-1alpha (fe. doi: 10.5664/jcsm.8682).

Response: Thank you very much for this interesting suggestion. We added a new paragraph referring to other possible pathways that you recommended, and inserted 2 more references (from line 283 to 289, page 7), including the reference you recommended:

“The bidirectional relation between OSA and liver is supported by high levels of HIF-1α (hypoxia-inducing factor), a serum protein and crucial transcription factor responsible for oxygen metabolism homeostasis [72]. Chronic intermittent hypoxia, seen in patients with OSA, increases the levels of HIF-1α in organs like brain and liver, aggravating the progression of NAFLD. HIF-1α promotes liver fibrosis in NAFLD by activating PTEN/p65 signalling pathway, which may be targeted for therapy [73]”.

With all our gratitude for your help and support,

The Authors

Round 2

Reviewer 1 Report

Manuscript ID: brainsci-1054771-revised

Title: Comprehensive overview of sleep disorders in patients with chronic liver disease

Journal: Brain Sciences

Introduction

  1. Page 2, lines 59-60. Authors wrote “We included in this systematic review 161 publications following PRISMA guidelines.”. The selection process of articles, according to PRISMA guidelines, should be graphically represented through a flow-chart.

Assessment of sleep disorders in patients with CLD

  1. Page 9, line 370. Authors should replace “SDs” with “sleep quality”.
  2. Page 10, lines 414-419. Since the Morningness-Eveningness Questionnaire does not measure sleep, this paragraph should be removed.
  3. Page 11, line 456. Authors should replace “Actigraphy” with “c) Actigraphy”.

Author Response

Journal: Brain Sciences

Manuscript: brainsci-1054771

Dear Reviewer,

We are very thankful for your notes, time and support in improving our paper; we have carefully read the comments and have revised/ completed the manuscript accordingly. Our responses are given in a point-by-point manner below (in blue), as well all the changes to the manuscript are highlighted in red.

In order to improve the quality of the manuscript, the text was modified, completed, and corrected, respecting exactly your recommendations. We hope that in the revised form, our manuscript will be found as suitable for publication.

Introduction

  1. Page 2, lines 59-60. Authors wrote “We included in this systematic review 161 publications following PRISMA guidelines.” The selection process of articles, according to PRISMA guidelines, should be graphically represented through a flow-chart.

We have added the flow chart, as the Reviewer suggested: Figure 1 (Page 2).

Assessment of sleep disorders in patients with CLD

   1. Page 9, line 370. Authors should replace “SDs” with “sleep quality”.

It was corrected as the Reviewer kindly requested (page 10, line 377)

   2. Page 10, lines 414-419. Since the Morningness-Eveningness Questionnaire does not measure sleep, this paragraph should be removed.

The paragraph from page 10, lines 414-419, was removed as the Reviewer kindly requested

   3. Page 11, line 456. Authors should replace “Actigraphy” with “c) Actigraphy”.

It was corrected as the Reviewer kindly requested (page 11, line 458)

With all our gratitude for your help and support,

The Authors

Reviewer 2 Report

Author's responded well to the comments.

One reference is unclear for me however. in sentence "The circadian clock misalignment is also encountered in OSA as a consequence of chronic sleep deprivation and hypoxia-reoxygenation, which have adverse effects on wake-promoting neurons in basal forebrain and brainstem. Besides, it has been shown that structural brain changes in white and grey matter are present in patients with OSA, leading to cognitive deficits and sleepiness [74]" (lines 292-296, page 7). The fragment describes circadian missalinment in OSA - the reference seems not a good match. Thre are few studies on the topic that should be referenced here (doi): 10.3390/jcm9051599,10.3389/fmed.2017.00187, 10.3390/jcm8101634).

Author Response

Journal: Brain Sciences

Manuscript: brainsci-1054771

Dear Reviewer,

We are very thankful for your notes, time and support in improving our paper; we have carefully read the comments and have revised/ completed the manuscript accordingly. Our responses are given in a point-by-point manner below (in blue), as well all the changes to the manuscript are highlighted in red.

In order to improve the quality of the manuscript, the text was modified, completed, and corrected, respecting exactly your recommendations. We hope that in the revised form, our manuscript will be found as suitable for publication.

One reference is unclear for me however. In sentence "The circadian clock misalignment is also encountered in OSA as a consequence of chronic sleep deprivation and hypoxia-reoxygenation, which have adverse effects on wake-promoting neurons in basal forebrain and brainstem. Besides, it has been shown that structural brain changes in white and grey matter are present in patients with OSA, leading to cognitive deficits and sleepiness [74]" (lines 292-296, page 7). The fragment describes circadian missalinment in OSA - the reference seems not a good match. Thre are few studies on the topic that should be referenced here (doi): 10.3390/jcm9051599,10.3389/fmed.2017.00187, 10.3390/jcm8101634).

We have added new sentences with information suggested from the 3 references and inserted them in the references list (page 22, lines 806-815).

Page 8, lines 295-303: “Gabryelska et al suggested in a recent study that patients with OSA are also at risk for developing clock disruption, a process which might be mediated by HIF-1α, since its increased level was associated with the overexpression of circadian clock proteins [75]. Previously to these findings, a group of researchers investigated the expression level of mRNA coding for clock genes. They reported that this level was altered in OSA patients compared to healthy controls and did not decrease after one month of CPAP treatment [76]. Consistent with the aforementioned results, Yang et al showed that the transcripts of all the investigated circadian clock genes displayed daily oscillation patterns in peripheral blood of controls, while 3 of them were arrhythmic in patients with OSA [77].”

With all our gratitude for your help and support,

The Authors